# Exploiting Excessive Invariance caused by Norm-Bounded Adversarial Robustness

## Abstract

Adversarial examples are malicious inputs crafted to cause a model to misclassify them. In their most common instantiation, "perturbation-based" adversarial examples introduce changes to the input that leave its true label unchanged, yet result in a different model prediction. Conversely, "invariance-based" adversarial examples insert changes to the input that leave the model's prediction unaffected despite the underlying input's label having changed. So far, the relationship between these two notions of adversarial examples has not been studied, we close this gap.

We demonstrate that solely achieving perturbation-based robustness is insufficient for complete adversarial robustness. Worse, we find that classifiers trained to be $\ell_p$-norm robust are *more* vulnerable to invariance-based adversarial examples than their undefended counterparts. We construct theoretical arguments and analytical examples to justify why this is the case. We then illustrate empirically that the consequences of excessive perturbation-robustness can be exploited to craft new attacks. Finally, we show how to attack a *provably robust* defense — certified on the MNIST test set to have at least $87\%$ accuracy (with respect to the original test labels) under perturbations of $\ell_\infty$-norm below $\varepsilon = 0.4$ — and reduce its accuracy (under this threat model with respect to an ensemble of human labelers) to $60\%$ with an automated attack, or just $12\%$ with human-crafted adversarial examples.

## 1 Introduction

Research on adversarial examples is motivated by a spectrum of questions. These range from the security of models deployed in the presence of real-world adversaries, to the need to capture limitations of representations and their (in)ability to generalize (Gilmer et al., 2018a). The broadest accepted definition of an adversarial example is "an input to a ML model that is intentionally designed by an attacker to fool the model into producing an incorrect output" (Goodfellow & Papernot, 2017).

Many formal definitions of adversarial examples were introduced since their initial discovery (Szegedy et al., 2013; Biggio et al., 2013). In a majority of work, adversarial examples are formalized as adding a perturbation $\delta$ to some test example $x$ to obtain an input $x^*$ that produces an incorrect model outcome.[1] We refer to this class of malicious inputs as *perturbation-based adversarial examples*.

To enable concrete progress, the adversary's capabilities may optionally be constrained by placing a bound on the maximum perturbation $\delta$ added to the original input. The goal of this constraint is to ensure that semantics of the input are left unaffected by the perturbation $\delta$. In the computer vision domain, $\ell_p$ norms have grown to be the default metric to measure semantic similarity. This led to a series of proposals for increasing the robustness of models to perturbation-based adversaries that operate within the constraints of an $\ell_p$ ball. These include robust optimization (Madry et al., 2017), explicit regularization of a model's Lipschitz constant (Cisse et al., 2017), or a variety of techniques to build models that are *provably robust* to small $\ell_p$ perturbations (Wong & Kolter, 2018; Raghunathan et al., 2018; Zhang et al., 2019).

In this paper, we show that optimizing a model's robustness to $l_p$-bounded perturbations is not only insufficient to address the lack of generalization identified via adversarial examples, but also potentially *harmful*. Intuitively, as $\ell_p$ distances are only crude approximations to the true visual

---

[1] Here, an incorrect output either refers to the model returning any class different from the original *source* class of the input, or a specific *target* class chosen by the adversary prior to searching for a perturbation.

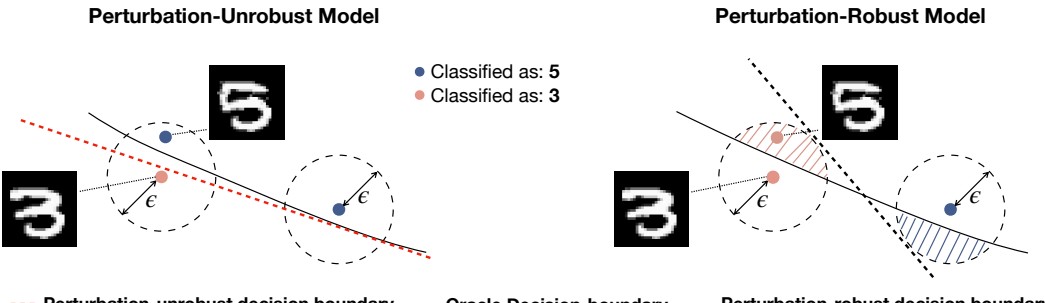

Figure 1: Real image: 3; Invariance-based attack: 5. [Left]: Training a classifier without constraints may learn a decision boundary unrobust to perturbation-based adversarial examples. [Right]: Enforcing robustness to norm-bounded perturbations introduces erroneous invariance (dashed regions in epsilon spheres). Note, that we display real data here, the misclassified 5 is an image found by our invariance-based attack which resides within a typically reported $\epsilon$-region around the displayed 3 (in the $\ell_0$ norm). This excessive invariance of the perturbation-robust model in task-relevant directions may be exploited, as shown by the attack proposed in this paper.

similarity in a given task, over-optimizing a model's robustness to $\ell_p$-bounded perturbations also renders the model invariant to actual semantics of the underlying task.

Excessive invariance of a model to class semantics can give rise to a new class of adversarial perturbations called *invariance adversarial examples* (Jacobsen et al., 2019). These are perturbations that successfully change the human-assigned ground-truth label of an input, while keeping the model's classification of the perturbed input unchanged (for example, we take an image of a '1' digit and perturb it—without changing the model's classification—until a human would agree that the digit now represents a '7'). See Figure 1 for an intuitive illustration of this phenomenon.

Whereas previous efforts (e.g., (Jacobsen et al., 2019)) explored different notions of adversarial examples independently, our work is the first to expose a complex relationship between perturbation-based and invariance-based adversarial examples.

Specifically, we introduce analytical constructions and empirical evidence that shows that increasing a model's robustness to perturbation-based adversarial examples can *increase* the model's vulnerability to invariance-based adversarial examples.

We formally show how to construct a model that is robust to perturbation-based adversarial examples but not to invariance-based adversarial examples. We further demonstrate how an imperfect model for the adversarial spheres task proposed by Gilmer et al. (2018b) is either vulnerable to perturbation-based or invariance-based attacks—depending on whether the point attacked is on the inner or outer sphere. Hence, these two types of adversarial examples are needed to fully account for model failures.

Finally, we empirically demonstrate with a new attack how an adversary can exploit the excessive invariance of certain models. We introduce the first algorithm that crafts invariance-based adversarial examples for the $\ell_0$ and $\ell_\infty$ norms, and show that many common models disagree with human labelers on these examples. In particular, our algorithm breaks a *provably-robust* defense on MNIST (Zhang et al., 2019).[2] This model is certified to have 87% test-accuracy under $l_\infty$-perturbations of radius $\varepsilon = 0.4$. Yet, on our automatically-generated invariance-based adversarial examples, the model only agrees with the human-assigned label in 60% of the cases; when we manually craft invariance adversarial examples we reduce accuracy to just 12% as determined by an ensemble of humans.

## 2    PERTURBATION- AND INVARIANCE-BASED ADVERSARIAL EXAMPLES

In order to make precise statements about adversarial examples, we begin with two definitions.

---

[2]Note that while MNIST is typically a poor choice for studying adversarial robustness (Carlini et al., 2019), in our case we explicitly chose to work with MNIST because it is the *only* dataset for which models have been claimed to be provably robust to non-negligible $l_p$ norm balls.

**Definition 1** (Perturbation-based Adversarial Examples). *Let $G$ denote the $i$-th layer, logit or argmax of the classifier. A **Perturbation-based adversarial example** (or perturbation adversarial) $x^* \in \mathbb{R}^d$ corresponding to a legitimate test input $x \in \mathbb{R}^d$ fulfills:*

1. *Created by adversary: $x^* \in \mathbb{R}^d$ is created by an algorithm $\mathcal{A} : \mathbb{R}^d \rightarrow \mathbb{R}^d$ with $x \mapsto x^*$.*

2. *Perturbation of output: $\|G(x^*) - G(x)\| > \delta$ and $\mathcal{O}(x^*) = \mathcal{O}(x)$, where perturbation $\delta > 0$ is set by the adversary and $\mathcal{O} : \mathbb{R}^d \rightarrow \{1, \ldots, C\}$ denotes the **oracle**.*

*Furthermore, $x^*$ is $\epsilon$-**bounded** if $\|x - x^*\| < \epsilon$, where $\| \cdot \|$ is a norm on $\mathbb{R}^d$ and $\epsilon > 0$.*

Property (i) allows us to distinguish perturbation adversarial examples from points that are misclassified by the model without adversarial intervention. Furthermore, the above definition incorporates also adversarial perturbations designed for hidden features as in (Sabour et al., 2016), while usually the decision of the classifier $D$ (argmax-operation on logits) is used as the perturbation target. Our definition also identifies $\epsilon$-bounded perturbation-based adversarial examples (Goodfellow et al., 2015) as a specific case of unbounded perturbation-based adversarial examples. However, our analysis primarily considers the latter, which correspond to the threat model of a stronger adversary.

**Definition 2** (Invariance-based Adversarial Examples). *Let $G$ denote the $i$-th layer, logit or argmax of the classifier. An **invariance-based adversarial example** (or invariance adversarial) $x^* \in \mathbb{R}^d$ corresponding to a legitimate test input $x \in \mathbb{R}^d$ fulfills:*

1. *Created by adversary: $x^* \in \mathbb{R}^d$ is created by an algorithm $\mathcal{A} : \mathbb{R}^d \rightarrow \mathbb{R}^d$ with $x \mapsto x^*$.*

2. *Lies in pre-image of $x$ under $G$: $G(x^*) = G(x)$ and $\mathcal{O}(x) \neq \mathcal{O}(x^*)$, where $\mathcal{O} : \mathbb{R}^d \rightarrow \{1, \ldots, C\}$ denotes the **oracle**.*

As a consequence, $D(x) = D(x^*)$ also holds for invariance-based adversarial examples, where $D$ is the output of the classifier. Intuitively, adversarial perturbations cause the output of the classifier to change, while the oracle would still label the new input $x^*$ in the original source class. Whereas perturbation-based adversarial examples exploit the classifier's *excessive sensitivity in task-irrelevant directions*, invariance-based adversarial examples explore the classifier's pre-image to identify *excessive invariance in task-relevant directions*: its prediction is unchanged while the oracle's output differs. Briefly put, perturbation-based and invariance-based adversarial examples are complementary failure modes of the learned classifier.

## 3 ROBUSTNESS TO PERTURBATION-BASED ADVERSARIAL EXAMPLES CAN CAUSE INVARIANCE-BASED VULNERABILITIES

We now investigate the relationship between the two adversarial example definitions from Section 2. So far, it has been unclear whether solving perturbation-based adversarial examples implies solving invariance-based adversarial examples, and vice versa. In the following, we show that this relationship is intricate and developing models robust in one of the two settings only would be insufficient.

In a general setting, invariance and stability can be uncoupled. For example, consider a linear classifier with matrix $A$. The perturbation-robustness is tightly related to forward stability (largest singular value of $A$). On the other hand, the invariance-view relates to the stability of the inverse (smallest singular value of $A$) and to the null-space of $A$. As largest and smallest singular values are uncoupled for general matrices $A$, the relationship between both viewpoints is likely non-trivial in practice.

### 3.1 BUILDING OUR INTUITION WITH EXTREME UNIFORM CONTINUITY

In the extreme, a classifier achieving perfect uniform continuity would be a constant classifier. Let $D : \mathbb{R}^n \rightarrow [0, 1]^C$ denote a classifier with $D(x) = y^*$ for all $x \in \mathbb{R}^d$. As the classifier maps all inputs to the same output $y^*$, there exist no $x^*$, such that $D(x) \neq D(x^*)$. Thus, the model is trivially perturbation-robust (at the expense of decreased utility). On the other hand, the pre-image of $y^*$ under $D$ is the entire input space, thus $D$ is arbitrarily vulnerable to invariance-based adversarial examples. Because this toy model is a constant function over the input domain, no perturbation of an initially correctly classified input can change its prediction.

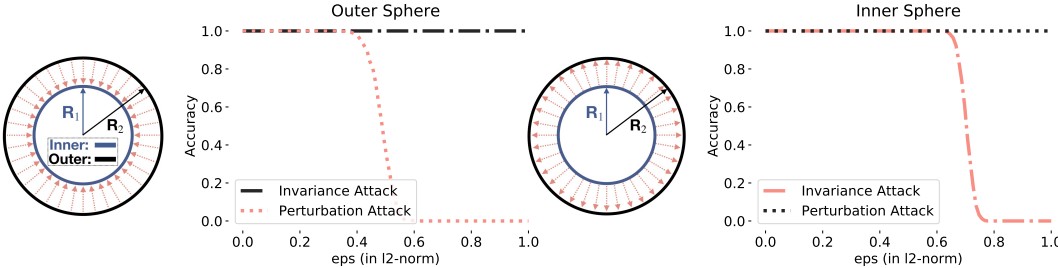

Figure 2: Robustness experiment on spheres with radii $R_1 = 1$ and $R_2 = 1.3$ and max-margin classifier that does not see $n = 10$ dimensions of the $d = 500$ dimensional input. [Left]: Attacking points from the outer sphere with perturbation-based attacks; accuracy drops when increasing the upper bound on $\ell_2$-norm perturbations. [Right]: Attacking points from the inner sphere with invariance-based attacks; accuracy drops when increasing the upper bound on $\ell_2$-norm perturbations. Each attack has a different effect on the manifold. Red arrows indicate the only possible direction of attack for each sphere. Perturbation attacks fail on the inner sphere, while invariance attacks fail on the outer sphere. Hence, both attacks are needed for a full account of model failures.

This trivial model illustrates how one not only needs to control *sensitivity* but also *invariance* alongside *accuracy* to obtain a robust model. Hence, we argue that the often-discussed tradeoff between accuracy and robustness (see Tsipras et al. (2019) for a recent treatment) should in fact take into account at least three notions: accuracy, sensitivity, and invariance. This is depicted in Figure 1. In the following, we present arguments as for why this insight can also extend to almost perfect classifiers.

### 3.2 COMPARING INVARIANCE-BASED AND PERTURBATION-BASED ROBUSTNESS

We now show how the analysis of perturbation-based and invariance-based adversarial examples can uncover different model failures. To do so, we consider the synthetic *adversarial spheres problem* of Gilmer et al. (2018b). The goal of this synthetic task is to distinguish points from two cocentric spheres (class 1: $\|x\|_2 = R_1$ and class 2: $\|x\|_2 = R_1$) with different radii $R_1$ and $R_2$. The dataset was designed such that a robust (max-margin) classifier can be formulated as:

$$D^*(x) = \text{sign}\left(\|x\|_2 - \frac{R_1 + R_2}{2}\right).$$

Our analysis considers a similar, but slightly sub-optimal classifier in order to study model failures in a controlled setting:

$$D(x) = \text{sign}\left(\|x_{1,\ldots,d-n}\|_2 - b\right),$$

which computes the norm of $x$ from its first $d - n$ cartesian-coordinates and outputs -1 (resp. +1) for the inner (resp. outer) sphere. The bias $b$ is chosen based on finite training set (see Appendix A).

Even though this sub-optimal classifier reaches nearly 100% on finite test data, the model is imperfect in the presence of adversaries that operate on the manifold (i.e., produce adversarial examples that remain on one of the two spheres but are misclassified). Most interestingly, the perturbation-based and invariance-based approaches uncover different failures (see Appendix A for details on the attacks):

- **Perturbation-based:** All points $x$ from the outer sphere (i.e., $\|x\|_2 = R_2$) can be perturbed to $x^*$, where $\mathcal{O}(x) = D(x) \neq D(x^*)$ while staying on the outer sphere (i.e., $\|x^*\|_2 = R_2$).

- **Invariance-based:** All points $x$ from the inner sphere ($\|x\|_2 = R_1$) can be perturbed to $x^*$, where $D(x) = D(x^*) \neq \mathcal{O}(x^*)$, despite being in fact on the outer sphere after the perturbation has been added (i.e., $\|x^*\|_2 = R_2$).

In Figure 2, we plot the mean accuracy over points sampled either from the inner or outer sphere, as a function of the norm of the adversarial manipulation added to create perturbation-based and invariance-based adversarial examples. This illustrates how the robustness regime differs significantly between the two variants of adversarial examples. Therefore, by looking only at perturbation-based

(respectively invariance-based) adversarial examples, important model failures may be overlooked. This is exacerbated when the data is sampled in an unbalanced fashion from the two spheres: the inner sphere is robust to perturbation adversarial examples while the outer sphere is robust to invariance adversarial examples (for accurate models).

# 4 INVARIANCE-BASED ATTACKS IN PRACTICE

We now show that our argument is not limited to the analysis of synthetic tasks, and give practical automated attack algorithms to generate invariance adversarial examples. We elect to study the only dataset for which robustness is considered to be nearly solved under the $\ell_p$ norm threat model: MNIST (Schott et al., 2019). We show that MNIST models trained to be robust to perturbation-based adversarial examples are *less* robust to invariance-based adversarial examples. As a result, we show that while *perturbation* adversarial examples may not exist within the $\ell_p$ ball around test examples, *adversarial examples* still do exist within the $\ell_p$ ball around test examples.

**Why MNIST?** The MNIST dataset is typically a poor choice of dataset for studying adversarial examples, and in particular defenses that are designed to mitigate them (Carlini et al., 2019). In large part this is due to the fact that MNIST is significantly different from other vision classification problems (e.g., features are quasi-binary and classes are well separated in most cases). However, the simplicity of MNIST is why studying $\ell_p$-norm adversarial examples was originally proposed as a toy task to benchmark models (Goodfellow et al., 2015). However, several years later, it is now argued that training MNIST classifiers whose decision is constant in an $\ell_p$-norm ball around their training data provides robustness to adversarial examples (Schott et al., 2019; Madry et al., 2017; Wong & Kolter, 2018; Raghunathan et al., 2018).

MNIST is the only dataset for which robustness to adversarial examples is considered even remotely close to being solved (Schott et al., 2019) and researchers working on (provable) robustness to adversarial examples have moved on to other, larger vision datasets such as CIFAR-10 (Madry et al., 2017; Wong et al., 2018) or ImageNet (Lecuyer et al., 2018; Cohen et al., 2019).

This section argues that, contrary to popular belief, MNIST is far from being solved. We show why robustness to $\ell_p$-norm perturbation-based adversaries is insufficient, even on MNIST, and why defenses with unreasonably high uniform continuity can harm the performance of the classifier and make it more vulnerable to other attacks exploiting this excessive invariance.

## 4.1 A TOY WORST-CASE: BINARIZED MNIST CLASSIFIER

To give an initial constructive example, consider a MNIST classifier which binarizes (by thresholding at, e.g., 0.5) all of its inputs before classifying them with a neural network. As (Tramèr et al., 2018; Schott et al., 2019) demonstrate, this binarizing classifier is highly $\ell_\infty$-robust, because most perturbations in the pixel space do not actually change the (thresholded) feature representation.

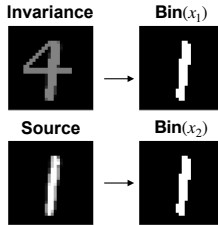

However, this binary classifier will have trivial invariance-based adversarial examples. Figure 8 shows an example of this attack. Two images which are dramatically different to a human (e.g., a digit of a one and a digit of a four) can become identical after pre-processing the images with a thresholding function at 0.5 (as examined by, e.g., Schott et al. (2019)).

Figure 3: Invariance-based adversarial example (top-left) is labeled differently by a human than original (bottom-left). However, both become identical after binarization.

With this toy worst-case discussed, we now turn to the full MNIST evaluation.

## 4.2 GENERATING MODEL-AGNOSTIC INVARIANCE-BASED ADVERSARIAL EXAMPLES

In the following, we build on existing invariance-based attacks (Jacobsen et al., 2019; Behrmann et al., 2018; Li et al., 2019) to propose a model-agnostic algorithm for crafting invariance-based adversarial examples. That is, our attack algorithm generates invariance adversarial examples that cause a human to change their classification, but where most models, not known by the attack algorithm, will *not* change their classification.

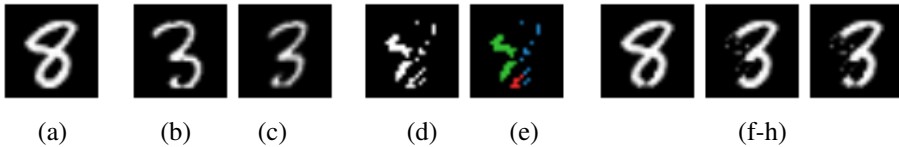

(a)  (b) (c)  (d) (e)   (f-h)

Figure 4: Process for generating $\ell_0$ invariant adversarial examples. From left to right: (a) the original image of an 8; (b) the nearest training image (labeled as 3), before alignment; (c) the nearest training image (still labeled as 3), after alignment; (d) the $\delta$ perturbation between the original and aligned training example; (e) spectral clustering of the perturbation $\delta$; and (f-h) possible invariance adversarial examples, selected by applying subsets of clusters of $\delta$ to the original image. (f) is a failed attempt at an invariance adversarial example. (g) is successful, but introduces a larger perturbation than necessary (adding pixels to the bottom of the 3). (h) is successful and minimally perturbed.

Our algorithm for generating invariance-based adversarial examples is simple, albeit tailored to work specifically on datasets where comparing images in pixel space is meaningful, like MNIST.

Begin with a *source* image, correctly classified by both the oracle evaluator (i.e., a human) and the model. Next, try all possible affine transformations of training data points whose label is different from the source image, and find the *target* training example which—once transformed—has the smallest distance to the source image. Finally, construct an invariance-based adversarial example by perturbing the source image to be "more similar" to the target image under the $\ell_p$ metric considered. In Appendix B, we describe instantiations of this algorithm for the $\ell_0$ and $\ell_\infty$ norms. Figure 4 visualizes the sub-steps for the $\ell_0$ attack, which are described in details in Appendix B.

The underlying assumption of this attack is that small affine transformations are *less likely* to cause an oracle classifier to change its label of the underlying digit than $\ell_p$ perturbations. In practice, we validate this hypothesis with a human study in Section 4.3.

Before evaluating this attack, we make one definition we will use to discuss the accuracy of

**Definition 3** (Successful $\epsilon$-bounded Invariance Attack). *For a given test example $x$ assigned label $y$, a* successful invariance attack *is a perturbed example $x^*$ such that the oracle classification $\mathcal{O}(x^*) \neq y$ and $\|x - x^*\| < \epsilon$, where $\|\cdot\|$ is a norm on $\mathbb{R}^d$ and $\epsilon > 0$*

In practice, we obtain the oracle classification by asking an ensemble of human labelers to label the point $x^*$; if more than some fraction of them agree on the label (throughout this section, 70%) and that label is different from the original, we call the attack successful. Note that success or failure is independent of any machine learning model: it only has to do with whether or not the underlying label has actually changed according to the (human) oracle.

## 4.3 EVALUATION

**Attack analysis.** We generate 1000 adversarial examples using each of the two above approaches on examples randomly drawn from the MNIST test set. Our attack is slow, with the alignment process taking (amortized) minutes per example. We performed no optimizations of this process and expect it could be improved. The mean $\ell_0$ distortion required is 25.9 (with a median of 25). The $\ell_\infty$ adversarial examples always use the full budget of $0.3$ or $0.4$ and take a similar amount of time to generate; most of the cost is again dominated by finding the nearest (transformed) training image.

**Human Study.** We randomly selected 100 examples from the MNIST test set and create 100 invariance-based adversarial examples under the $\ell_0$ norm and $\ell_\infty$ norm, as described above. We then conduct a human study to evaluate whether or not these invariance adversarial examples indeed are successful as defined earlier, i.e., whether humans agree that the label has been changed. We additionally hand-crafted 50 invariance adversarial examples under the specific norms. The process to create these examples was quite simple: we built a minimal image editor that enabled us change images at a pixel level under a given $\ell_p$ constraint. One author then modified 50 random test examples in the way that they perceived as changing the underlying class. We presented 40 human evaluators with these 100 images, half of which were natural unmodified MNIST digits, and the remaining half were distributed randomly between $\ell_0$ or $\ell_\infty$ invariance adversarial examples.

| Attack Type | Success Rate |
|---|---|
| Clean Images | 0% |
| $\ell_0$ Attack | 55% |
| $\ell_\infty, \varepsilon = 0.3$ Attack | 21% |
| $\ell_\infty, \varepsilon = 0.4$ Attack | 37% |

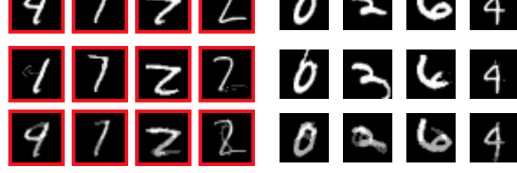

(a) Success rate of our invariance adversarial example causing humans to switch their classification.

(b) Original test images (top) with our $\ell_0$ (middle) and $\ell_\infty$ (bottom) invariance adversarial examples. (left) successful attacks; (right) failed attacks.

Figure 5: Our invariance-based adversarial examples. Humans (acting as the oracle) switch their classification of the image from the original test label to a different label.

**Results.** For the clean (unmodified) test images, 98 of the 100 examples were labeled correctly by *all* human evaluators. The other 2 images were labeled correctly by over 90% of human evaluators.

Our $\ell_0$ attack is highly effective: For 55 of the 100 examples at least 70% of human evaluator who saw that digit assigned it the same way, with a different label from the original test label. Humans only agreed with the original test label (with the same 70% threshold) on 34 of the images, while they did not form a consensus on the remaining 18 examples. The (much simpler) $\ell_\infty$ attack is less effective: with a distortion of 0.3 the oracle label changed 21% of the time and with 0.4 the oracle label changed 37% of the time. We summarize results in Table 5 (a).

In Figure 5 (b) we show sample invariance adversarial examples. To simplify the analysis in the following section, we split our generated invariance adversarial examples into two sets: the successes and the failures, as determined by whether the plurality decision by humans was different than or equal to the human label. We only evaluate the models on the subset of invariance adversarial examples that caused the humans to switch their classification.

**Model Evaluation.** Given oracle ground-truth labels for each of the images (as decided by humans), we report how often models agree with the human-assigned label. Table 1 summarizes this analysis. For the invariance adversarial examples we report model accuracy only on the *successful* attacks.

Every classifier labeled all successful $\ell_\infty$ invariance adversarial examples **incorrectly** (with one exception where the $\ell_2$ PGD-trained classifier Madry et al. (2017) labeled one of the invariance adversarial examples correctly). Despite this fact, PGD adversarial training and Analysis by Synthesis Schott et al. (2019) are two of the state-of-the-art $\ell_\infty$ perturbation-robust classifiers.

The situation is more complex for the $\ell_0$-invariance adversarial examples. In this setting, the models which achieve *higher* $\ell_0$ perturbation-robustness result in *lower* accuracy on this new invariance test set. For example, Bafna et al. (2018) develops a $\ell_0$ perturbation-robust classifier that relies on the sparse Fourier transform. This perturbation-robust classifier is substantially weaker to invariance adversarial examples, getting only 38% accuracy compared to a baseline classifier's 54% accuracy.

**Breaking Certified Defenses.** Our invariance attacks are sufficiently strong that they constitute a *break* of some certified defenses. For example, Zhang et al. (2019) develop a certified defense to $\ell_\infty$ adversarial examples which *proves* that the accuracy on the test set when perturbed by $\varepsilon = 0.4$ is at least 87%. When we run their pre-trained model on all 100 of our $\varepsilon = 0.4$ invariance adversarial examples we find it has a 96% "accuracy" (i.e., it matches the original test label 96% of the time). However, when we compare the accuracy of this model compared to the new labels as assigned by an ensemble of humans, the accuracy is just 63%.[3] That is, while the proof in the paper is *mathematically correct* it does not actually deliver 87% accuracy on any new test input perturbed by $\varepsilon = 0.4$: humans would have changed their classification in many of these settings. Worse, for the 50 adversarial examples we crafted by hand, the model *disagrees* with the human ensemble 88% of the time: it has just 12% accuracy.

---

[3]For all invariance adversarial examples the most likely label was selected by more than half of humans. If we sub-set to only the 21 examples where *all* humans agreed on the label, the accuracy of this model remains at 50%; if instead we require at least 75% agreement the accuracy is 65%.

| Agreement between model and humans, for *successful* invariance adversarial examples | | | | | | | |
|---|---|---|---|---|---|---|---|
| **Model:** [a] | **ResNet** | **CNN** | **$\ell_0$ Sparse** | **Binary-ABS** | **ABS** | **$\ell_\infty$ PGD** | **$\ell_2$ PGD** |
| Clean | 99% | 99% | 99% | 99% | 99% | 99% | 99% |
| $\ell_0$ | 65% | 54% | 38% | 47% | 58% | 56%* | 27%* |
| $\ell_\infty, \varepsilon = 0.3$ | 0% | 0% | 0%* | 0% | 0% | 0% | 5%* |

Table 1: Model accuracy with respect to the oracle human labelers on the subset of examples where the human-obtained oracle label is different from the test label. Models which are more robust to *perturbation* adversarial examples (such as those trained with adversarial training) agree with humans **less often** on *invariance-based* adversarial examples. Values denoted with an asterisks * violate the perturbation threat model of the defense and should not be taken to be attacks. When the model is *wrong*, it classified the input as the original test label, and not the new oracle label.

[a]$\ell_0$ Sparse: Bafna et al. (2018); ABS and binary-ABS: Schott et al. (2019); $\ell_\infty$ PGD and $\ell_2$ PGD: Madry et al. (2017)

## 4.4 NATURAL IMAGES

While the previous discussion focused on synthetic (Adversarial Spheres) and simple tasks like MNIST, similar phenomena may arise in natural images. In Figure 6, we show two different $\ell_2$ perturbations of the original image (left). The perturbation of the middle image is nearly imperceptible and thus the classifier's decision should be robust to such changes. On the other hand, the image on the right went through a semantic change (from tennis ball to a strawberry) and thus the classifier should be sensitive to such changes (even though this case is ambiguous due to two objects in the image). However, in terms of the $\ell_2$ norm the change in the right image is even smaller than the imperceptible change in the middle. Hence, making the classifier robust within this $\ell_2$ norm-ball will make the classifier vulnerable to invariance-based adversarial examples like the semantic changes in the right image.

A similar argument can be made for other norms. For instance Co et al. (2018) show that a perturbation of magnitude 16/255 in $\ell_\infty$, can suffice to give an image of a cat the appearance to be printed on a shower curtain (both are classes in Imagenet). However, we leave studying this direction in more detail to future work.

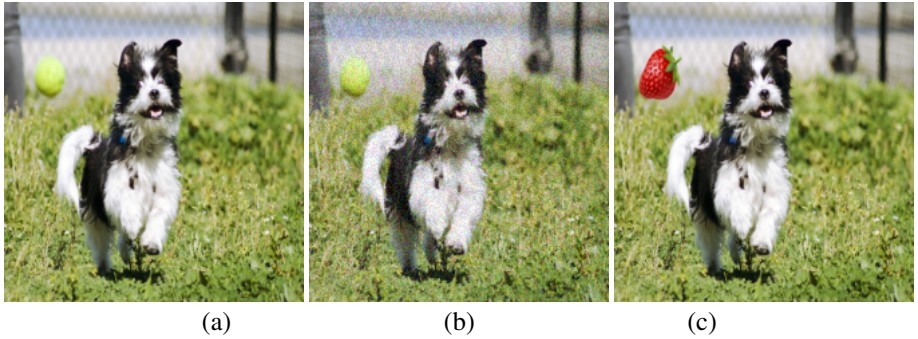

(a)        (b)        (c)

Figure 6: Visualization that large $\ell_2$ norms can also fail to measure semantic changes in images. (a) original image in the ImageNet test set labeled as a *tennis ball*; (b) imperceptible perturbation, $\ell_2 = 24.3$; (c) semantic perturbation with a $\ell_2$ perturbation of 23.2 that removes the tennis ball.

## 5 CONCLUSIONS

Training models robust to perturbation-based adversarial examples should not be treated as equivalent to learning models robust to *all* adversarial examples. While most of the research has focused on perturbation-based adversarial examples that exploit excessive classifier *sensitivity*, we show that the reverse viewpoint of excessive classifier *invariance* should also be taken into account when evaluating

robustness. Furthermore, other unknown types of adversarial examples may exist: it remains unclear whether or not the union of perturbation and invariance adversarial examples completely captures the full space of evasion attacks.

**Consequences for $\ell_p$-norm evaluation and certified defenses.**  Our invariance-based attacks are able to find (non-perturbation-based) adversarial examples within the $\ell_p$ ball on classifiers that were trained to be robust to $\ell_p$-norm perturbation-based adversaries. As a consequence of this analysis, researchers should carefully set the radii of $\ell_p$-balls when measuring robustness to norm-bounded perturbation-based adversarial examples. Robustness to small epsilons can indeed lead to a better trade-off between perturbation- and invariance-based robustness (Engstrom et al., 2019). However, due to the misalignment between p-norms with the "true perceptual distance metric" of the oracle, training against larger epsilon (as is common on MNIST for instance, and we expect will soon become true on larger datasets) will inevitably lead to erroneous invariances and hence to vulnerability with respect to invariance-based adversarial examples. Furthermore, setting a consistent radius across all of the data may be difficult: we find in our experiments that some class pairs are more easily attacked than others by invariance-based adversaries.

In particular, we find that recent proposals that claim exceptionally high $\ell_0$ and $\ell_\infty$ norm-bounded robustness are not meaningfully correct. Given the claimed perturbation budget, it is possible to generate invariance adversarial examples that fool these defenses. Indeed, another by-product of our study is to showcase the importance of human studies when the true label of candidate adversarial inputs becomes ambiguous and cannot be inferred algorithmically.

Defenses to adversarial examples on MNIST must stop blindly increasing the distortion threshold and claiming robustness, and in particular for $\ell_\infty$ distortion of $\varepsilon = 0.4$. Any claim of robustness greater than $60\%$ accuracy is not meaningfully correct because at this distortion level, it is possible to generate examples where humans change their classification.

**Invariance.**  Our work confirms findings reported recently in that it surfaces the need for mitigating undesired invariance in classifiers. The cross-entropy loss as well as architectural elements such as ReLU activation functions have been put forward as possible sources of excessive invariance (Jacobsen et al., 2019; Behrmann et al., 2018). Other work has pointed out a trade-off as a consequence of robustness to perturbations in different frequency spectra (Yin et al., 2019).

However, more work is needed to develop quantitative metrics for invariance-based robustness. One promising architecture class to control invariance-based robustness are invertible networks (Dinh et al., 2014) because, by construction, they cannot build up any invariance until the final layer (Jacobsen et al., 2018; Behrmann et al., 2019).

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

## A  DETAILS ABOUT ADVERSARIAL SPHERES EXPERIMENT

In this section, we provide details about the Adversarial Spheres (Gilmer et al., 2018b) experiment. First, the bias $b$ is chosen, such that the classifier $D$ is the max-margin classifier on the (finite) training set $\mathcal{X}$ (assuming separability: $l \leq u$):

$$l = \max_{\|x\|_2 = R_1, x \in T} \|x_{1,\ldots,d-n}\|_2, \quad u = \min_{\|x\|_2 = R_2, x \in T} \|x_{1,\ldots,d-n}\|_2, \quad b = l + \frac{u-l}{2}.$$

Second, the attacks are designed such that the adversarial examples $x^*$ stay on the data manifold (two concentric spheres). In particular, following steps are taken:

**Perturbation-based:** All points $x$ from the outer sphere (i.e., $\|x\|_2 = R_2$) can be perturbed to $x^*$, where $\mathcal{O}(x) = D(x) \neq D(x^*)$, while staying on the outer sphere (i.e., $\|x^*\|_2 = R_2$) via following steps:

1. Perturbation of decision: $x^*_{1,\ldots,d-n} = a\,(x_{1,\ldots,d-n})$, where scaling $a > 0$ is chosen such that $\|x^*_{1,\ldots,d-n}\|_2 < b$

2. Projection to outer sphere: $x^*_{d-n,\ldots,d} = c\,(x_{d-n,\ldots,d})$, where scaling $c > 0$ is chosen such that $\|x^*_{d-n,\ldots,d}\|_2 = \sqrt{R_2^2 - \|x^*_{1,\ldots,d-n}\|_2^2}$

For points $x$ from the inner sphere, this is not possible if $b > R_1$.

**Invariance-based:** All points $x$ from the inner sphere ($\|x\|_2 = R_1$) can be perturbed to $x^*$, where $D(x) = D(x^*) \neq \mathcal{O}(x^*)$, despite being in fact on the outer sphere after the perturbation has been added (i.e., $\|x^*\|_2 = R_2$) via following steps:

1. Fixing the used dimensions: $x^*_{1,\ldots,d-n} = x_{1,\ldots,d-n}$

2. Perturbation of unused dimensions: $x^*_{d-n,\ldots,d} = a\,(x_{d-n,\ldots,d})$, where scaling $a > 0$ is chosen such that $\|x^*_{d-n,\ldots,d}\|_2 = \sqrt{R_2^2 - \|x^*_{1,\ldots,d-n}\|_2^2}$

For points $x$ from the outer sphere, this is not possible if $b > R_1$.

## B  DETAILS ABOUT MODEL-AGNOSTIC INVARIANCE-BASED ATTACKS

Here, we give details about our model-agnostic invariance-based adversarial attacks on MNIST.

**Generating $\ell_0$-invariant adversarial examples.** Assume we are given a training set $\mathcal{X}$ consisting of labeled example pairs $(x, y)$. As input our algorithm accepts an example $\hat{x}$ with oracle label $\mathcal{O}(\hat{x}) = \hat{y}$. Image $\hat{x}$ with label $\hat{y} = 8$ is given in Figure 4 (a).

Define $\mathcal{S} = \{x : (x, y) \in \mathcal{X}, y \neq \hat{y}\}$, the set of training examples with a different label. Now we define $\mathcal{T}$ to be the set of transformations that we allow: rotations by up to 20 degrees, horizontal or vertical shifts by up to 6 pixels (out of 28), shears by up to $20\%$, and re-sizing by up to $50\%$.

Now, we generate the new augmented training set $\mathcal{X}^* = \{(t(x), y, t) : t \in \mathcal{T}, (x, y) \in \mathcal{X}\}$. By assumption, each of these examples is labeled correctly by the oracle. In our experiments, we verify the validity of this assumption through a human study and omit any candidate adversarial example that violates this assumption. Finally, we search for

$$x^*, y^*, t = \underset{(x^*, y^*, t) \in \S^*}{\arg\min} \|x^* - \hat{x}\|_0.$$

By construction, we know that $\hat{x}$ and $x^*$ are similar in pixel space but have a different label. Figure 4 (b-c) show this step of the process. Next, we introduce a number of refinements to make $x^*$ be "more similar" to $\hat{x}$. This reduces the $\ell_0$ distortion introduced to create an invariance-based adversarial example—compared to directly returning $x^*$ as the adversarial example.

First, we define $\delta = |\hat{x} - x^*| > 0.5$ where the absolute value and comparison operator are taken element-wise. Intuitively, $\delta$ represents the pixels that substantially change between $x^*$ and $\hat{x}$. We choose $0.5$ as an arbitrary threshold representing how much a pixel changes before we consider the change "important". This step is shown in Figure 4 (d). Along with $\delta$ containing the *useful* changes that are responsible for changing the oracle class label of $\hat{x}$, it also contains irrelevant changes that are superficial and do not contribute to changing the oracle class label. For example, in Figure 4 (d) notice that the green cluster is the only semantically important change; both the red and blue changes are not necessary.

To identify and remove the superficial changes, we perform spectral clustering on $\delta$. We compute $\delta_i$ by enumerating all possible subsets of clusters of pixel regions. This gives us many possible **potential** adversarial examples $x_i^* = \hat{x} + \delta_i$. Notice these are only potential because we may not actually have applied the necessary change that actually changes the class label.

We show three of the eight possible candidates in Figure 4. In order to alleviate the need for human inspection of each candidate $x_i^*$ to determine which of these potential adversarial examples is actually misclassified, we follow an approach from Defense-GAN Samangouei et al. (2018) and the Robust Manifold Defense Ilyas et al. (2017): we take the generator from a GAN and use it to assign a likelihood score to the image. We make one small refinement, and use an AC-GAN Mirza & Osindero (2014) and compute the class-conditional likelihood of this image occurring. This process reduces $\ell_0$ distortion by $50\%$ on average.

As a small refinement, we find that initially filtering $\mathcal{X}$ by $20\%$ least-canonical examples makes the attack succeed more often.

**Generating $\ell_\infty$-invariant adversarial examples.** Our approach for generating $\ell_\infty$-invariant examples follows similar ideas as for the $\ell_0$ case, but is conceptually simpler as the perturbation budget can be applied independently for each pixel (as we will see, our $\ell_\infty$ attack is however less effective than the $\ell_0$ one, so further optimizations may prove useful).

We build an augmented training set $\mathcal{X}^*$ as in the $\ell_0$ case. Instead of looking for the closest nearest neighbor for some example $\hat{x}$ with label $\mathcal{O}(\hat{x}) = \hat{y}$, we restrict our search to examples $(x^*, y^*, t) \in \mathcal{X}^*$ with specific target labels $y^*$, which we've empirically found to produce more convincing examples (e.g., we always match digits representing a 1, with a target digit representing either a 7 or a 4). We then simply apply an $\ell_\infty$-bounded perturbation (with $\epsilon = 0.3$) to $\hat{x}$ by interpolating with $x^*$, so as to minimize the distance between $\hat{x}$ and the chosen target example $x^*$.

## C   INVARIANCE-BASED ADVERSARIAL EXAMPLES FOR BINARIZED MNIST

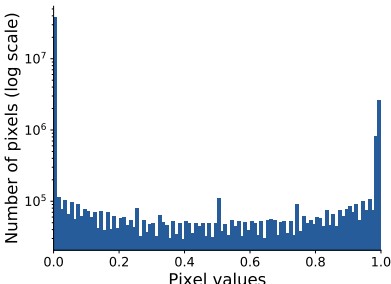

Figure 7: Histogram of MNIST pixel values (note the log scale on the y-axis) with two modes around $0$ and $1$. Hence, binarizing inputs to a MNIST model does not impact its performance importantly.

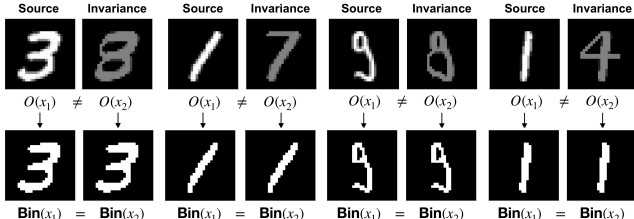

Figure 8: Invariance-based adversarial examples for a toy $\ell_\infty$-robust model on MNIST. By thresholding inputs, the model is robust to perturbations $\delta$ such that $\|\delta\|_\infty \lesssim 0.5$. Adversarial examples (top-right of each set of 4 images) are labeled differently by a human. However, they become identical after binarization; the model thus labels both images confidently in the source image's class.

## D   COMPLETE SET OF 100 INVARIANCE ADVERSARIAL EXAMPLES

Below we give the 100 randomly-selected test images along with the invariance adversarial examples that were shown during the human study.

### D.1 ORIGINAL IMAGES

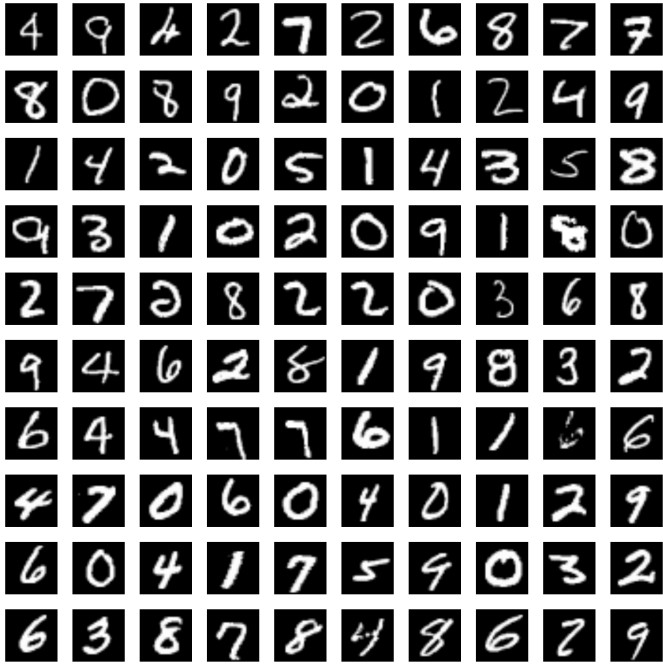

### D.2 $\ell_0$ INVARIANCE ADVERSARIAL EXAMPLES

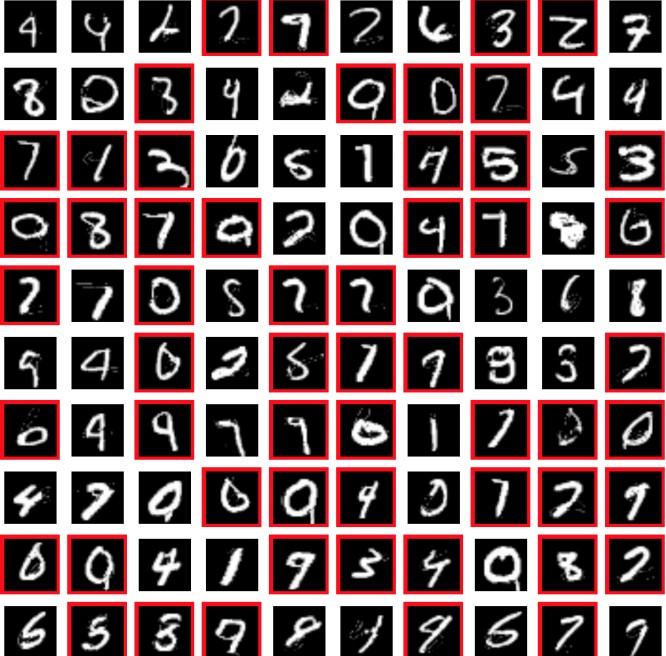

## D.3 $\ell_\infty$ INVARIANCE ADVERSARIAL EXAMPLES

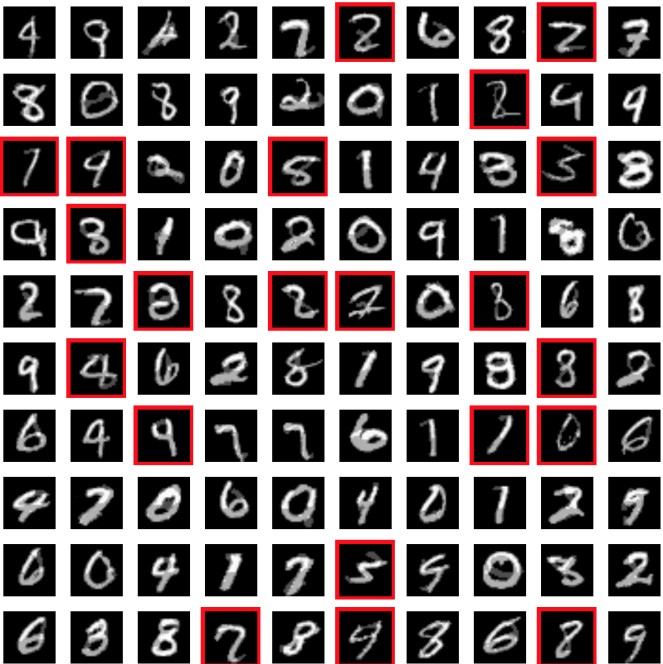

