# OpenReview forum: "Exploiting Excessive Invariance caused by Norm-Bounded Adversarial Robustness"
_ICLR.cc/2020/Conference — Reject_

### Official Review · AnonReviewer2 · 2019-10-22
**Official Blind Review #2**

**Rating:** 6

**Review:**

Thanks for your response. It has addressed most of my concerns. I'd like to increase my score.

-----------------

This paper examines the perturbation-based and invariance-based robustness of deep neural networks. They found that models trained to be robust under l_p threat model are more vulnerable to invariance-based adversarial examples. This paper provides several illustrative examples to tell the reasons behind this phenomenon. An attack method for generating invariance-based adversarial examples is also proposed to attack the (provably robust) models.

Overall, this paper analyzes the invariance-based robustness of deep neural networks with norm-bounded robustness, which is an interesting problem. Several illustrative examples provide reasons on why models robust to perturbation-based adversarial examples could be more sensitive to invariance-based adversarial examples.

However, my main concern is on the contributions of this paper. A clear contribution of the paper is to study the perturbation-based and invariance-based robustness together, and present several demo examples to illustrate that. Are there any other contributions in this paper? Is the proposed attack method novel compared with previous methods? And what are the differences between the proposed attack with previous methods? The authors could discuss more on the attack method and compare its performance with others.

I'd like to raise my currently rating based on the author feedback.

**Experience Assessment:**

I have read many papers in this area.

**Review Assessment: Checking Correctness Of Derivations And Theory:**

I carefully checked the derivations and theory.

**Review Assessment: Checking Correctness Of Experiments:**

I carefully checked the experiments.

**Review Assessment: Thoroughness In Paper Reading:**

I read the paper at least twice and used my best judgement in assessing the paper.

---

> ### Author Response · Authors · 2019-11-15
> **Response Blind Review #2**
>
> Thank you very much for your review.
>
> We will summarize our main contributions here again and updated the draft to make sure they are more prominent in the manuscript.
>
> We have 3 major contributions:
>
> 1) Exposing an fundamental trade-off between norm-bounded perturbation- and invariance-based robustness.
>
> We argue based on toy examples and theoretical constructions that p-norm robustness is not only insufficient for adversarial robustness, but will inevitably increase vulnerability to invariance-based adversaries.
>
> 2) Introducing a novel invariance-based adversarial attack.
>
> We introduce an attack that exploits excessive invariance of deep networks and use it to attack defended and undefended classifiers. This attack is new and the first of its kind. There are three invariance-based attacks in the literature to the best of our knowledge (see references in the first sentence of section 4.2). Two rely on pre-image search via sophisticated techniques and another one is analytical and relies on invertible networks. All of them do not allow to flexibly craft invariance-based attacks with respect to a specified norm, which we require here. Hence, our attack is the first that enables us to highlight how classifier more robust to norm-bounded adversarial examples are *less* robust to this attack than their undefended counterparts.
>
> 3) Showing certified robustness with currently achievable epsilon values becomes harmful under the face of invariance-based adversaries.
>
> We attack the state-of-the-art in certified robustness with our *model-agnostic* invariance-based adversarial examples and show that we can exploit excessive invariance by definition. Therefore, we manage to reduce the certified accuracy on the original test set from 87% to 60% and 12% after applying our attack. This shows that our analysis already has practical implications. Increasing the distortion bounds for norm-bounded robustness hence introduces new vulnerabilities.

---

### Official Review · AnonReviewer3 · 2019-10-23
**Official Blind Review #3**

**Rating:** 3

**Review:**


================ Update after reading the rebuttal and the revised paper =======================================

I have now read the author rebuttal and the revised paper. I had raised two main issues with the paper in my initial review (see below): 1) experiments don't provide enough support for the main claim; 2) the main claim seems to contradict an earlier result by Engstrom et al. (2019).

With respect to the second point, the authors have responded that Engstrom et al. (2019) use relatively small perturbation sizes during adversarial training and that the trade-off between invariance vs. perturbation based adversarial robustness may be different for such small perturbation sizes. This sounds plausible, although I would have much preferred a more direct demonstration of this (for example, by adversarially training MNIST models with different perturbation sizes, then looking at their performance on invariance-based adversarial examples).

With respect to the first point, the authors have pointed to some numbers in Table 1, but my criticism still stands: for example, the perturbation-robust ABS performs better than the undefended CNN model (similarly for the l_inf PGD model, even though it is trained for a different metric). This seems to contradict the authors' claim that excessive perturbation robustness leads to worse performance on invariance-based adversarial examples.

In conclusion, I very much appreciate the authors' rebuttal and I definitely think there are some interesting ideas in this paper. However, given that one of my main concerns has not been addressed and the other one only insufficiently addressed, I'm inclined to keep my score as it is. Going forward, I would encourage the authors to develop the trade-off between invariance vs. perturbation based adversarial sensitivity for different perturbation sizes more thoroughly.

========================================================================================================

This paper argues that the standard notion of l_p-norm bounded adversarial examples does not adequately capture all types of adversarial examples we may care about. In particular, the authors note the existence of adversarial examples caused by the excessive invariance of classifiers to semantically meaningful perturbations. This was noted in earlier works as well, but the current paper purports to establish a link between l_p-ball robustness and invariance-based adversarial examples. It also introduces a method to generate such invariance-based adversarial examples. There are interesting ideas in this paper, however I have some questions and concerns about some of the claims made in the paper that I would like to see addressed. Here are the main issues for me:

1) If I’m not mistaken, the main claim of the paper, namely that l_p-ball robustness worsens the performance on invariance-based adversarial examples does not actually seem to be supported by any result in the paper beyond some of the simplest, toyest examples.  For example, in the MNIST examples in Table 1, I don’t see how l_p-ball robust models are performing worse than the other models on the invariance-based adversarial examples (the results seem mixed at best). Even in some of the toy examples, this claim is not supported: for example, in the adversarial spheres example, the l_p-ball robust classifier would correspond to the max-margin classifier, which is not vulnerable to invariance-based adversarial examples. Even the ad-hoc sub-optimal classifier chosen in this example doesn't show that l_p-ball robust models are more vulnerable to invariance-based adversarial examples, just that the two phenomena are distinct. There’s the single ImageNet example in Figure 6, which is cute and is consistent with the claim, however it’s just a single example. Also please note that this example does not work with the l_inf norm, so it just shows the inadequacy of the l_2 norm, not of l_p norms in general.

2) The main claim of this paper seems to contradict a result in Engstrom et al. (2019) (https://arxiv.org/abs/1906.00945). They also demonstrate the existence of invariance-based adversarial examples in non-robust models (e.g. see their Figure 2), but their results seem to suggest that these types of examples do not arise in robust models (models trained with adversarial training). Please discuss this paper and clarify the seeming discrepancy between the results there and your claims.

Other issues:

3) It is really hard to follow the invariance-based adversarial example generation process described in section 4.2. Please describe this more clearly, motivating each step of the process (currently the steps seem ad hoc, it is not explained why each step is needed), so that a reader can understand the generation process at a high level without having to consult the appendix.

4) The labels are misaligned in Figure 4.

5) In section 4.3, it is mentioned that “we additionally hand-crafted 50 invariance adversarial examples under the specific norms”. It is not explained at all why these additional examples are needed and it is not described how exactly these are generated (neither in the main text, nor in the appendix as far as I can see).

6) In Table 1, the different models are not described at all (neither in the main text, nor in the appendix). They are just acronyms right now. Please describe what these models are.

7) The idea that there may be trade-offs between different notions of robustness and that l_p-ball robustness is not the be all and end all of all robustness measures have been noted in some previous works: for example, Yin et al. (2019): https://arxiv.org/abs/1906.08988 Although the phenomenon discussed in Yin et al. (2019) is different from the one highlighted in the current paper, the main message seems quite similar. So, please discuss this connection in your paper.

**Experience Assessment:**

I have published one or two papers in this area.

**Review Assessment: Checking Correctness Of Derivations And Theory:**

N/A

**Review Assessment: Checking Correctness Of Experiments:**

I carefully checked the experiments.

**Review Assessment: Thoroughness In Paper Reading:**

I read the paper at least twice and used my best judgement in assessing the paper.

---

> ### Author Response · Authors · 2019-11-15
> **Response Blind Review #3**
>
> Thank you very much for your review. We have updated the manuscript based on your feedback and believe it helped to greatly improve important points.
>
> Q: l_p-ball robustness worsens the performance on invariance-based adversarial examples does not actually seem to be supported
> ================
> A: Table 1 shows that in the L0 metric, the undefended ResNet classifier performs significantly better than all defended ones. The CNN without residual connections also performs better than the models defended in the L0 threat model (L0-sparse and binary ABS). Furthermore, we illustrate how binarization (which is used by the SOTA binary-ABS model) increases vulnerability to invariance-based adversarial examples also for the L-infinity norm. This is explained in Figure 3 and section 4.1.
>
> The adversarial spheres classifier is indeed different, as an optimally robust max-margin solution exists. Its purpose is not to show that there is a trade-off in this optimal case, but to show what you mention as well: the phenomena are distinct and we miss out vulnerabilities if we only consider one viewpoint.
>
> Q: ImageNet example is consistent with claim but only valid for L2
> ================
> A: Similar arguments can be made for other norms. For instance [1] show that a perturbation of magnitude 16/255 in Linfinity, can suffice to give various Imagenet images the appearance to be printed on a shower curtain (shower curtain is a class in Imagenet). However, we leave studying this direction in more detail to future work.
>
> [1] Co, Kenneth T., et al. "Procedural Noise Adversarial Examples for Black-Box Attacks on Deep Convolutional Networks." (2018).
>
> Q: The main claim of this paper seems to contradict a result in Engstrom et al. (2019)
> ================
> A: Thanks for bringing this up, we have discussed the paper in the updated manuscript (in conclusion section). The results of Engstrom et al. do not contradict ours. They simply correspond to different tradeoffs between robustness and Invariance for different perturbation magnitudes.
> Engstrom et al. consider very small perturbations (near imperceptible). Thus, the fact that adversarial training reduces existence of invariance-based adversarial examples in this setting may very well be true. However, due to the misalignment between p-norms with the “true” perceptual distance metric of the oracle, training against larger epsilon (as is common on MNIST for instance, and we expect will soon become true on larger datasets) will inevitably lead to erroneous invariances and hence to vulnerability with respect to invariance-based adversarial examples. Avoiding this would mean to choose a smaller epsilon at the expense of vulnerability to perturbation-based adversaries (see figure 6). The only way adversarial training would never run into this problem is to use the correct perceptual distance metric. However, if we would have access to this metric, we wouldn’t need to train machine learning models but simply use nearest neighbor search.
>
> Q: Please describe adversarial example generation process more clearly
> ================
> A: We gave a high-level description of the algorithm in the beginning of section 4.2 including an illustrative example with step-by-step instruction in figure 4. We intentionally moved the technical details to the appendix for brevity. We would be happy to further revise our manuscript to clarify in the main body of the text any part of the attack that remains unclear.
>
> Q: Labels are misaligned in figure 4
> ================
> A: We have fixed this.
>
> Q: Explain why handcrafted examples are needed and how they are generated
> ================
> A: We resorted to manual creation of invariant adversarial examples mainly because automatically generating convincing examples is challenging. We added a human into the loop to create a larger set of Linf adversarial examples without having to make any simplifying assumptions.
> Intuitively, this makes sense as we're trying to create small perturbations that maximally change a human's classification. So having a human-in-the-loop for the generation process helps in creating more convincing examples.
> The process to create these examples was quite simple: we built a minimal image editor that enabled us change images at a pixel level under a given lp constraint. One author then modified 50 random test examples in the way that they perceived as changing the underlying class. We then validated these examples in a human study: other humans agreed that the label had indeed changed, even though the perturbations were under the infinity-norm constraint.
> Developing a fully automated process for crafting convincing invariance adversarial examples that does not suffer from the same limitations than existing perturbation-based adversarial examples is an interesting (albeit challenging) avenue for future work.

---

> > ### Author Response · Authors · 2019-11-15
> > **Part II**
> >
> > Q: Citations for methods in table 1
> > ================
> > A: We apologize for the oversight and have corrected this; the models are as follows:
> > L0 sparse refers to Bafna et al. 2018 at NeurIPS (“Thwarting Adversarial Examples: An L0-Robust Sparse Fourier Transform”),
> > ABS and Binary-ABS refers to Schott et al. 2018 at ICLR (“Towards the first adversarially robust neural network model on MNIST”),
> > L_infinity PGD and L_2 PGD refer to Madry et al. 2017 at ICLR (“Towards Deep Learning Models Resistant to Adversarial Attacks”)
> >
> > Q: Relationship to Yin et al. (2019)
> > ================
> > A: The reference Yin et al. (2019) is indeed conceptually similar as it presents a robustness tradeoff between two types of corruptions (low- vs. high-frequency perturbations). Very much in line with our work, they show how hardening the model towards one type of perturbation results in a less robust model with respect to another kind of perturbation. As their results on data augmentation show, considering a broader set of perturbation types increase the robustness of the model.
> > Thus, we think Yin et al. (2019) supports our viewpoint of broadening the study of adversarial examples since they observe a similar behavior as we do (however in a different setting: frequency perturbations vs. lp invariance/perturbation attacks).

---

### Official Review · AnonReviewer1 · 2019-10-24
**Official Blind Review #1**

**Rating:** 6

**Review:**

This paper proposes a fine-grained definition for adversarial robustness, dividing adversarial robustness into perturbation robustness and invariance robustness. Same as the previous definition of adversarial robustness (robustness against imperceptible perturbations), perturbation robustness reflects the model's ability to maintain the prediction after a label-preserving transform. Invariance robustness, on the other hand, reflects the model's flexibility against invariance-based adversaries which changes the actual label within the norm ball around clean samples. With examples from the two-sphere experiment, MNIST dataset with large epsilons and a triplet of natural images, it reaches the conclusion that training models to be invariant to any sample within a norm ball is a bad idea when epsilon is too large.

I like this paper in that it points out the problems with some established evaluation protocols for adversarial robustness, e.g., evaluating the robustness on MNIST with a max l_infty norm of 0.4. However, it comes at no surprise that large epsilon gives rise to non-label-preserving perturbations on images. No automated solution can be inspired from the paper. A more valuable direction would be to evaluate the minimum l_p distance between classes, but it seems intractable at the moment.

Also, it should be emphasized that the invariance-based adversarial examples exist only when epsilon is improperly high for image classification tasks. This can also be verified by the two-sphere experiments. By taking average of the accuracy on the inner sphere and outer sphere, the accuracy against perturbation attack drops before invariance attack with the increasing epsilon, demonstrating invariance attack is only a problem when epsilon is too large for training the model. For other tasks such as language understanding, invariance-based adversarial examples may be a much severe problem for the seemingly robust models.

As a conclusion, I tend to accept this paper to draw more attention to a better notion of robustness, or developing more sophisticated approaches to defending against perturbation and invariance attacks simultaneously at a larger epsilon. However, the current version is still short of any foreseeable solution. Still, this is perhaps the best we can hope for.


**Experience Assessment:**

I have published one or two papers in this area.

**Review Assessment: Checking Correctness Of Derivations And Theory:**

I assessed the sensibility of the derivations and theory.

**Review Assessment: Checking Correctness Of Experiments:**

I assessed the sensibility of the experiments.

**Review Assessment: Thoroughness In Paper Reading:**

I made a quick assessment of this paper.

---

> ### Author Response · Authors · 2019-11-15
> **Response Blind Review #1**
>
> Thank you for your positive review.
>
> The main goal of our work is to raise awareness of a range of problems with p-norm bounded perturbation robustness. As you mentioned as well, the issues we describe are natural once pointed out, but so far have not been reported in the literature.
>
> As a consequence, multiple results and goals of adversarial robustness research are questionable. The most striking that the certified robustness *proof* of eps=0.4 in L_inf norm on MNIST is essentially vacuous: while true, this model performs worse on invariance adversarial examples and thus renders the guarantee meaningless as far as the security of the model is concerned.
>
> Unfortunately, as you point out as well, there is currently no solution to the problem. But we believe that a first step towards a solution is clearly laying out the problem and demonstrating that it persists in practice. This is the purpose of our paper.
>
> Right now, the only way to solve the problem once and for all would be to use a distance metric that corresponds exactly to human perception. However, this is of course a chicken and egg problem, because once we would have found this metric training machine learning models becomes meaningless: a nearest neighbor classifier would suffice.
>
> We hope that our work inspires researchers to state claims of robustness more carefully when studying adversarial robustness solely from the lp-bounded perturbation perspective. Our results should be understood as a word of caution to research solely aiming at increasing the perturbation bounds as this approach is bound to introduce new vulnerabilities in most realistic scenarios that would matter for practical security.

---

### Author Response · Authors · 2019-11-15
**Revision Summary**

We thank all the reviewers for their work, their insightful feedback and for considering our work to be interesting.

We have:

=> Clarified multiple points concerning our main contributions
=> Added and discussed additional requested references
=> Incorporated changes all over the manuscript to address as many concerns as possible

---

### Decision · Program_Chairs · 2019-12-19

**Decision:**

Reject

**Comment:**

The paper considers the relationship betwee:

- perturbations to an input x which change predictions of a model but not the ground truth label
- perturbations to an input x which do not change a model's prediction but do chance the ground truth label.

The authors show that achieving robustness to the former need not guarantee robustness to the latter.

While these ideas are interesting, the reviewers would like to see a tighter connection between the two forms of robustness developed.